# The impact of unemployment benefits on birth outcomes: Quasi-experimental evidence from European linked register data

**Dorian Kessler** *, **Debra Hevenstone**

Social Work Department, Bern University of Applied Sciences, Bern, Switzerland

* dorian.kessler@bfh.ch

## Abstract

Cash transfers have been shown to improve birth outcomes by improving maternal nutrition, increasing healthcare use, and reducing stress. Most of the evidence focuses on programs targeting the poorest in the US—a context with non-universal access to healthcare and strong health inequalities. It is thus unclear whether these results would apply to cash transfers targeting a less disadvantaged population and whether they apply to other contexts. We provide evidence on the impact of unemployment benefits on birth outcomes in Switzerland, where access to healthcare is near-universal and social assistance is relatively generous. Our study taps into a policy reform that reduced unemployment benefits by 56%. We use linked parent-child register data and difference-in-differences estimates as well as within sibling comparisons. We find that the reform did not impact birth outcomes when fathers were unemployed but reduced the birthweight of children when mothers were unemployed by 80g and body length by 6mm. There are stronger effects for children whose mothers were the primary earner before job loss, but effects do not differ systematically by household income. These results suggest that in the Swiss context, unemployment benefits improve birth outcomes by reducing (job search) stress rather than by improving nutrition or healthcare use. As such, cash transfers likely play a role for newborn health in most other contexts.

## Introduction

Through its negative effects on multiple life domains, job loss is a major source of stress [1]. While this stress is associated with deteriorated mental health [2–4], research generally suggests that overall physical health changes little in immediate response to unemployment [1, 3, 5–7]. With the exception of birth outcomes: Children who are *in utero* during their parents' unemployment are born lighter [8–10]. Lower birth weight, in turn, is linked to life-long disadvantages in health and economic opportunities [11–15].

The effect of job loss on childbirth outcomes likely stems from maternal stress, increased smoking and drinking, and insufficient nutrition and perinatal healthcare access [9, 16]. Social insurance and assistance programs are designed to counteract these negative effects, helping

**Data Availability Statement:** The data in this article are based on merged secondary individual register data owned by the Swiss government (unemployment (AVAM), population (STATPOP),

civil status (BEVNAT) and social security income registers (Individualkonten AHV)) and is classified as highly sensitive. All variables were anonymized before transmission to authors (e.g. social security numbers were pseudonymized). However, in combination, data would possibly allow for identification of individuals. Statistical calculations were performed without written consent by study subjects as Swiss regulations allow for the use of merged individual data without written consent if used for statistical purposes only. The authors have gained access to this data based on the data sharing contract nr. 180231 between the Swiss Federation and Bern University of Applied Sciences. Due to the nature of the data, contractual agreements prohibit data sharing with third parties. Data can be accessed by third parties via request to the Swiss Federal Statistical Office (Email to verknuepfungen@bfs.admin.ch using the form provided on https://www.bfs.admin.ch/bfsstatic/dam/assets/17084398/master; see https://www.bfs.admin.ch/bfs/en/home/services/data-linkages/for-third-parties.html for more information). Authors share data sharing contract, data access application form, and code for data preparation and analysis upon request to the corresponding author.

**Funding:** This research was funded by the Swiss National Science Foundation (grant nr. 176371) and Bern University of Applied Sciences.

**Competing interests:** The authors have declared that no competing interests exist.

individuals maintain consumption and reduce financial distress. Targeted in-kind assistance such as the US food stamp program (SNAP) [17] and the Special Supplemental Nutrition Program for Women, Family and Children (WIC) [18] have been shown to reduce pre-term birth and to increase birth weight. Currie and Cole [19] also found that social assistance (Aid to Families with Dependent Children, now called Temporary Assistance for Needy Families) increased birth weight. Examining changes in the Earned Income Tax Credit (EITC), Hoynes et al. [20] found that a $1,000 increase in cash transfers reduced low weight births by 2 to 3 percentage points. These studies provide evidence that social benefits targeted at lower income households can improve birth outcomes.

However, the current literature is limited in that it focuses on the US context and on social programs targeting the very poorest [17–20]. Social insurance, particularly in Europe, assists a broader population in a context of near-universal health insurance coverage. It is unclear whether cash transfers might also improve birth outcomes for less disadvantaged populations in contexts with broader access to healthcare.

In one recent US study, Noghanibehambari and Salari [21] looked at Unemployment Insurance (UI), a type of social insurance targeting a much broader population. They found that increasing UI generosity improved birth outcomes. The paper, however, had two methodological limitations. First, concurrent state-level trends in UI generosity and birth outcomes could drive results. Second, like other recent US studies on the effects of changes in state UI benefit generosity [22, 23], Noghanibehambari and Salari [21] were unable to exactly identify UI recipients in their data, but had to rely on imputed UI receipt based on mothers' race and education. Beyond methodological limitations, results from the US, with high health inequalities, birth outcomes on par with middle-income countries [11], recent downward trajectories in newborn health [24], and a substantial uninsured population, might not be applicable to other developed countries [25]. It is possible that in contexts with less health inequality and universal access to perinatal healthcare UI might not impact birth outcomes [26].

In this study, we provide evidence on the effect of cash transfers on birth outcomes (birth weight and body length at birth) for a broader population, focusing on a Swiss reform that reduced maximum UI benefit duration for the long-term unemployed by six months. The reform led to a loss of 56% of UI benefits among those reaching one year of unemployment. To measure the effects of the reform, we draw on merged administrative data and calculate difference-in-differences estimates. We compare pre-to-post reform changes in birth outcomes among newborns of unemployed parents affected by the reform to changes in birth outcomes among newborns unaffected by the reform. To account for selection bias, we also calculate triple differences, comparing the gap between newborns' outcomes and their closest older sibling, again comparing pre to post reform changes for groups affected vs. unaffected by the reform.

Building on the best existing studies on the effects of public cash transfers on birth outcomes, this is the first study combining a quasi-experimental research design with merged parent-child register data to measure the effect of cash transfers on birth outcomes. With this approach, we can provide reliable evidence on whether cash transfers impact newborns' health. The data also allows us to test differences in the effects of UI by household income and the relative income contribution of the unemployed parent prior to job loss. This subgroup analysis allows us to say something about the mechanisms underlying the effect of UI. For instance, finding that effects of UI also exist among more affluent households that do not rely on UI benefits to maintain basic consumption would speak against UI affecting newborn health through less or lower quality nutrition or healthcare usage, but through other mechanisms such as stress. Hence, our results not only speak to the literature on socio-economic

determinants of newborn health [16], but also to the study of economic shocks, stress and health in general [27, 28].

Given that Switzerland has generous social assistance [29] and health insurance schemes, finding effects in this context would suggest that cash transfers are key in protecting newborn health in any context. Such finding would be particularly relevant in the current context as many countries' UI benefit systems are in flux with years of benefits cuts [30], followed by expansion during the Covid-19 pandemic, and with countries now considering whether and to what extent they will retain expansions.

## Determinants of birth outcomes

Fetal development and birth outcomes such as birth weight are determined by genetic factors, maternal physiological and health conditions, health behaviors, environmental factors and stress [31, 32]. Birth weight is reduced for lower order and multiple births [31] and for children of young or old mothers [32]. Relevant health related risk factors are unhealthy diets [33], adverse health conditions during pregnancy such as influenza [34], inadequate pre-natal healthcare [31], smoking [35, 36] or drinking [37]. Relevant environmental factors are *in utero* exposure to agricultural [38, 39] or industrial [31] contamination. The role of stress for fetal development is evidenced by studies on the impact of cortisol [40] and stressful events such as natural disasters [15, 41] or the death of a relative [12].

## Unemployment and birth outcomes

On the one hand, unemployment can be expected to affect fetal development through stress that results from the reduced financial well-being, perceived pressure to find a new job, changes in daily routines and self-perception that follow job loss [1, 40]. Financial difficulties have been shown to lead to more negative health behaviors such as smoking and drinking [9, 20, 42]. On the other hand, the impact of unemployment on newborn health could run through less purchasing power [9]. If couples depend on the benefit income of the unemployed partner to afford healthy nutrition and adequate pre-natal care, the loss of that income could cause a less healthy diet or underuse of healthcare during critical periods of fetal development.

## Unemployment insurance in Switzerland and the 2011 reform

UI systems can be described by various characteristics, including eligibility requirements, income replacement levels, and potential benefit duration [43]. In Switzerland, eligibility depends on reasons for job loss, willingness to work, and paid UI contributions. Benefits are paid for both involuntary and voluntary job losses, but there is a waiting period in the latter case. Recipients must write a minimum number of job applications per month and participate in employment programs (except for women in gestation week 30 or later). Individuals must have a minimum of 1 year of contributions in the two years before claiming benefits. Replacement levels are stable throughout unemployment spells at 70% without dependents and 80% with. Benefits are capped at an annual salary of CHF 148'200. In terms of eligibility requirements and replacement levels, Swiss UI can be classified as generous compared to other OECD countries [30].

Potential benefit duration was lowered in a reform introduced in April 2011. The biggest sub-population affected were unemployed aged 25 to 53 with incomplete contribution histories (paid employment subject to UI contributions in the two years before unemployment). Unemployed that claimed UI benefits between July 2003 and March 2009 and who had 12–17 months of contributions were entitled to a maximum of 1.5 years of benefits, while with the same contribution history, such individuals would be entitled to a maximum of only one year

of benefits after the reform [44]. In contrast, unemployed aged 25 to 53 who paid UI contributions for a minimum of 18 months were entitled to 1.5 years of benefits, both before and after the reform. Benefits can be claimed with interruptions (e.g., due to temporary work) but are viable only within the first 24 months after unemployment start.

## The expected effects of the reform on birth outcomes and effect heterogeneity

Reduced unemployment benefits could have deteriorated birth outcomes through two main mechanisms. The loss of benefits for those who did not find a job before exceeding one year of benefit receipt is likely to have resulted in a) greater stress [45] and associated adverse health behaviors [42] as well as b) less positive consumption patterns [20], which both impact fetal development and consequently worsened birth outcomes.

It is likely that the effect of the reform varied with the economic household situation [46]. In households with sufficient alternative economic resources, the loss of benefits might not require drastic cuts in consumption that could affect quality of nutrition or pre-natal healthcare. In contrast, without alternative income or wealth, the loss in benefits causes more financial difficulties which are likely associated with stronger increases in fetal stress exposure. In principle, Switzerland has a relatively generous healthcare system with universal insurance coverage and no health insurance deductibles charged for pre-natal healthcare. Also, social assistance—the social safety net of last resort—is relatively generous and would allow for a healthy diet even without UI benefits. Still, we do not exclude the possibility that for poor households, a loss of UI benefits could also lead to less healthy diets and forgoing of relevant healthcare which would contribute to more negative effects of the reform.

Hence, we expect that the reform had a stronger effect on a) households that already had a low income before one of the parents became unemployed and b) households where the unemployed parent contributed a larger share to the households' total income before losing his or her job, i.e. where the household depended more crucially on the income of the unemployed parent. Finally, one might conjecture that lost UI income has a stronger impact on birth outcomes if the unemployed parent is the mother, given empirical evidence that income is not perfectly shared in households [47]. The loss of benefits by mothers is thus likely more directly related to fetal exposure to adverse developmental influences.

## Empirical approach

### Study design

To assess the effect of the reform, we set up a repeated cross-sectional *difference-in-differences* study design [48]. We assessed the difference in birth outcomes from the period before (unemployment starts between July 2003 and March 2009) to the period after reform (unemployment starts between October 2010 and August 2016) among the children of unemployed parents who, due to limited UI contributions (between 12 to 17 months of UI contributions in the two years before unemployment), were eligible for 1.5 years of benefits *before*, but only one year of benefits *after*, the reform. The reform created *increased* stress for these parents between the moment they *expected* to exhaust benefits (about 9 months after the onset of unemployment) until the moment they would have lost benefit entitlements irrespective of the reform (24 months after the onset of unemployment). We included all children of parents receiving *at least* 9 months of UI benefits and whose gestation began between months 9 and 23 following the onset of unemployment. These are the children whose fetal development we expect to have been affected by the reform (henceforth *treated children*). Robustness checks in the

supplementary information section show that the main conclusions do not change with slight adjustments in this time window (cf. S2 and S3 Tables).

To account for general trends in birth outcomes, we compared the change in newborns' outcomes for the *treated children* with the change in birth outcomes among a control group of newborns whose unemployed parents were unaffected by the reform but who also had inter-rupted contribution histories (18–23 months of contributions in the 24 months before unem-ployment and thus 1.5 years of benefit eligibility *both* before and after the reform, henceforth *control children*). As with the treated children, control children are also restricted to those whose parents had received benefits during at least 9 months and to those whose gestation began between 9 and 23 months following unemployment start.

Control children differ from the treatment group as their parents had more stable employ-ment histories, which is likely correlated with better parental health and fetal development. If such selectivity changed over calendar time, estimates using a control group with longer con-tributions could be biased. To account for such potential biases, we additionally calculate effects on changes in birth outcomes to last siblings (henceforth *preceding siblings)*. This focus on within-siblings change in birth outcomes has the advantage that unobserved but constant differences between treated and control children are removed from the analysis, reducing the problem of selection [8, 10]. However, the within-sibling approach has the disadvantage of restricting the analysis to second or higher order births as well as singleton births, reducing statistical power. Using both estimation strategies allows us a larger sample using the first and a more conservative confirmation of overall effects using the second.

## Data

Data was constructed by merging diverse individual-level administrative data sources. In Swit-zerland, ethical review boards do not need to approve social science studies using anonymized data for statistical purposes. Legal approval for the use of the data was obtained from the Swiss Federal Statistical Office and legally secured in the data sharing agreement between the Swiss Federation, FORS, the Canton of Bern and the Bern University of Applied Sciences (contract nr. 180231).

Information on unemployment is drawn from unemployment insurance registers (UIR) [49]. UIR provided information on unemployment spell start and number of months of UI contributions, allowing us to define the treated and control samples. Moreover, it included information on pre-unemployment educational attainment, occupation, income, and working hours of the unemployed parent, allowing us to account for pre-to-post-reform trends in these characteristics. UIR was merged onto national live birth registers to gather information on birth outcomes (birth weight, body length), birth characteristics (date, parity, sex, singleton status) as well as information on both parents (citizenship, age, civil status) [32, 50]. Because birth register data included social security identification numbers (SSI) only for births in 2010 or later, for earlier births we probabilistically linked birth registers to 2010 population registers [51] using unique combinations of maternal birth dates, paternal birth dates, marriage dates, and childbirth dates as pseudo-identifiers, excluding cases with non-unique combinations. Using this method, we were able to identify SSI of both parents for 74% of all births registered in Switzerland between 2003 and 2009.

We measured income for the entire study period using social security income registers (SSIR) which includes earned income from employment and self-employment as well as bene-fit income from unemployment, disability, motherhood and military insurance [52]. House-hold income is constructed as the sum of all income sources received by the unemployed individual and the child's other parent in the year before unemployment, with all income

inflation-adjusted to December 2014 levels. The relative income contribution of the unemployed parent is calculated as the percentage of his/her income in the household income.

## Study samples and treatment intensity

Our sample consists of 17'684 births, excluding the 3.8% of observations that were dropped due to missing data. Table 1 presents descriptive information including the mean values and percentage shares of all observed characteristics of treated and controls, both in the pre- and the post-reform period with estimated difference-in-differences (DID) ($\beta_{DID}$) obtained from a multiplicative interaction term between the binary variable *Treated* (versus control children) and the binary variable *Post* (versus pre-reform) in OLS models with each of the dimensions in Table 1 as the outcome variable [53].

The basic model is the following:

$$Outcome = \alpha + \beta_{Treated} * Treated + \beta_{Post} * Post + \beta_{DID} * Treated * Post$$
$$+ \beta_{Control\ variables} * Control\ variables + \epsilon \tag{1}$$

Table 1 shows that reform reduced the share of parents that received UI benefits by 23 percentage points in the second year of unemployment or, in monetary terms, by an average amount of CHF 840/month or 56% of the UI benefits they would have received without the reform. DID coefficients for the control variables show that characteristics of treated and control observations have different trajectories from the pre to the post period. Relative to the control sample, unemployed parents in the treated sample became more socioeconomically disadvantaged, as can be seen from the significant coefficients on pre-unemployment household income, occupation, and education. We account for these differential changes in observed characteristics of treated and control in our regression analyses (cf. next section).

## Analytical strategy

The main analysis starts with OLS models predicting birth outcomes. To assess the effects of the reform, we estimate $\beta_{DID}$ from Eq 1 accounting for covariate imbalance between comparison groups by including all control variables as linear controls in the models. We found that qualitative conclusions from our results do not depend on whether estimates are adjusted or not (cf. S1 Table). Our outcome measures are birth weight measured in grams and body length measured in centimeters. Our estimates of within-sibling change in birth outcomes is identical to Eq 1, except instead of levels, our outcome is the difference in birth outcomes between the focal child ($S_1$) and the preceding sibling ($S_0$).

$$Outcome_{S_1-S_0} = \alpha + \beta_{Treated} * Treated + \beta_{Post} * Post + \beta_{DID} * Treated * Post +$$
$$+ \beta_{Control\ variables} * Control\ variables + \epsilon \tag{2}$$

In our examination of heterogeneous treatment effects, we compare the impact of the reform by gender of the unemployed parent, household income, and the relative income contribution of the unemployed parent before job loss. To assess effect differences by household income, we repeat the calculation of $\beta_{DID}$ by household income terciles. To assess the role of relative income contribution, we distinguish three groups: unemployed parents who contributed less than one third of the household income ("secondary earners"), unemployed parents who contributed at least one third but less than two thirds ("egalitarian couples") and unemployed who contributed at least two thirds of household income ("primary earners"). We repeat the calculation of $\beta_{DID}$ for each of these groups.

**Table 1. Treatment intensity and sample characteristics.**

| | Controls, pre | Controls, post | Treated, pre | Treated, post | DiD estimate |
|---|---|---|---|---|---|
| **Treatment variables** | | | | | |
| UI benefits (%) | 61 | 62 | 61 | 38 | -23 *** |
| UI benefits (mean, CHF/month) | 1323 | 1513 | 1309 | 659 | -840 *** |
| **Control variables** | | | | | |
| Unemployed mother (%) | 54 | 52 | 45 | 41 | -2 |
| Income before unemployment (mean, CHF/month) | 3531 | 3741 | 2562 | 2611 | -161 * |
| Hours before unemployment (mean, 100% = 42hrs/wk) | 9172 | 9128 | 9224 | 9038 | -142 ** |
| Household income before unemployment (mean, CHF/month) | 8151 | 8792 | 6651 | 6859 | -432 ** |
| Married (%) | 86 | 78 | 84 | 78 | 1 |
| Occupation: managerial/professional (%) | 18 | 19 | 14 | 14 | -2 |
| Intermediate (%) | 49 | 47 | 43 | 40 | -1 |
| Manual (%) | 33 | 33 | 43 | 47 | 3 * |
| Education: tertiary (%) | 19 | 25 | 19 | 19 | -5 *** |
| Vocational (%) | 52 | 48 | 47 | 45 | 1 |
| Compulsory (%) | 29 | 27 | 34 | 35 | 4 ** |
| Age mother (mean) | 31 | 32 | 31 | 31 | 0 |
| Age father (mean) | 34 | 35 | 35 | 34 | -1 *** |
| Swiss citizenship mother (%) | 48 | 47 | 40 | 40 | 2 |
| Swiss citizenship father (%) | 44 | 41 | 37 | 32 | -2 |
| Singleton birth (%) | 99 | 96 | 99 | 96 | 0 |
| Parity | 2 | 2 | 2 | 2 | 0 |
| Region: Leman (%) | 27 | 28 | 33 | 31 | -4 ** |
| Mittelland (%) | 18 | 20 | 20 | 21 | 0 |
| Northwest (%) | 13 | 13 | 12 | 12 | -1 |
| Zurich (%) | 19 | 18 | 17 | 17 | 1 |
| East (%) | 11 | 11 | 8 | 10 | 2 |
| Central (%) | 7 | 6 | 5 | 5 | 0 |
| Ticino (%) | 5 | 4 | 5 | 5 | 2 * |
| **N** | **4081** | **6277** | **3635** | **3691** | **17684** |

Sample: Children whose first month of gestation was between month 9 and 23 after unemployment start. Parents with at least 9 months of unemployment. Treated: 12 to 17 months with UI contributions. Controls: 18 to 23 months of UI contributions. Pre unemployment start July 2003-March 2009. Post unemployment start October 2010-August 2016. UI benefit receipt measured in the second year of unemployment. Pre-unemployment incomes measured in the year before unemployment. P-value thresholds DID:

* = 5%,

** = 1%,

*** = 0,1%.

## Results

### Overall

Table 2 presents our DID estimates from the models looking at overall effects. Birth weight and body length changed between the samples of parents that were unemployed before the reform and the sample of parents that were unemployed after the reform, but in different directions for treated versus control children. We find a reduction in average birth weight ($\beta_{DID}$: -24g) that is not statistically significant and a reduction in body length due to the reform that is statistically significant ($\beta_{DID}$: -.2 cm, p < .01). When looking at birth outcomes

**Table 2. Birth outcomes controls and treated, pre- and post-reform means, difference-in-differences estimates, and 95% confidence intervals of the effect of the reform.**

| | Controls, pre | Controls, post | Treated, pre | Treated, post | DiD estimate |
|---|---|---|---|---|---|
| **Level** | | | | | |
| Birth weight (g) | 3308.3 | 3289.9 | 3321 | 3270.5 | -23.6 (-55.8;8.6) |
| Body length (cm) | 49.3 | 49.3 | 49.4 | 49.2 | -0.2 ** (-0.4;-0.1) |
| N | 4081 | 6277 | 3635 | 3691 | 17684 |
| **Difference to preceding sibling** | | | | | |
| Birth weight (g) | 48.9 | 79.4 | 89.2 | 65.1 | -52.7 + (-109;3.7) |
| Body length (cm) | 0.1 | 0.2 | 0.3 | 0.1 | -0.2 (-0.5;0.1) |
| N | 1562 | 5221 | 792 | 1494 | 9069 |

Sample: Children whose first month of gestation was between month 9 and 23 after unemployment start. Parents with at least 9 months of unemployment. Treated: 12 to 17 months with UI contributions. Controls: 18 to 23 months of UI contributions. Pre unemployment start July 2003-March 2009. Post unemployment start October 2010-August 2016. DiD estimates are adjusted for control variables listed in Table 1. P-value thresholds DID:

+ = 10%,

* = 5%,

** = 1%,

*** = 0,1%.

compared to preceding siblings, estimates of the effect of the reform on average birth weight are stronger in magnitude and marginally statistically significant ($\beta_{DID}$: -51g, p < .1). Estimates on body length are similar in magnitude but are not statistically significant.

## Heterogeneous effects

Fig 1 presents the results stratified by gender of the unemployed parent, household income before unemployment and the relative income contribution of the unemployed parent to household income before unemployment. The left-hand panel shows the effects of the reform are concentrated among unemployed mothers. For unemployed mothers the reform reduced average birth weight (main sample: -53g, p < .05, difference to preceding sibling: -112g, p < .01) and body length (main sample: -0.5 cm, p < .01, difference to preceding sibling: -0.7 cm, p < .01). The reform had no significant effect on the birth outcomes of children with unemployed fathers.

As we did not find any effects of the reform on the birth outcomes of children of unemployed fathers, we continued the analyses of household income and relative household income contribution focusing on unemployed mothers. The middle panel of Fig 1 shows that there is no pattern of differences in the reform's effect by household income tercile, except for slightly greater reductions in body length found for children of mothers in bottom tercile income households. However, we find robust evidence for differences depending on the relative income contribution of the unemployed mother to the household made before unemployment. While there are clearly no effects for mothers in equal earning households, reductions in birth weight and body length are clearly greatest for mothers who were primary earners before unemployment. The width of confidence intervals shows that this is a very small group, but effects are nevertheless significant at p < .05 for body length (both estimation strategies) and for the birth weight p>.05 using the difference to preceding siblings and p < .1 using levels.

## Interpretation and conclusions

Recent research suggests that the physical health of the unemployed remains remarkably stable after job loss [1, 3, 5–7]. However, the financial and non-financial stressors associated with

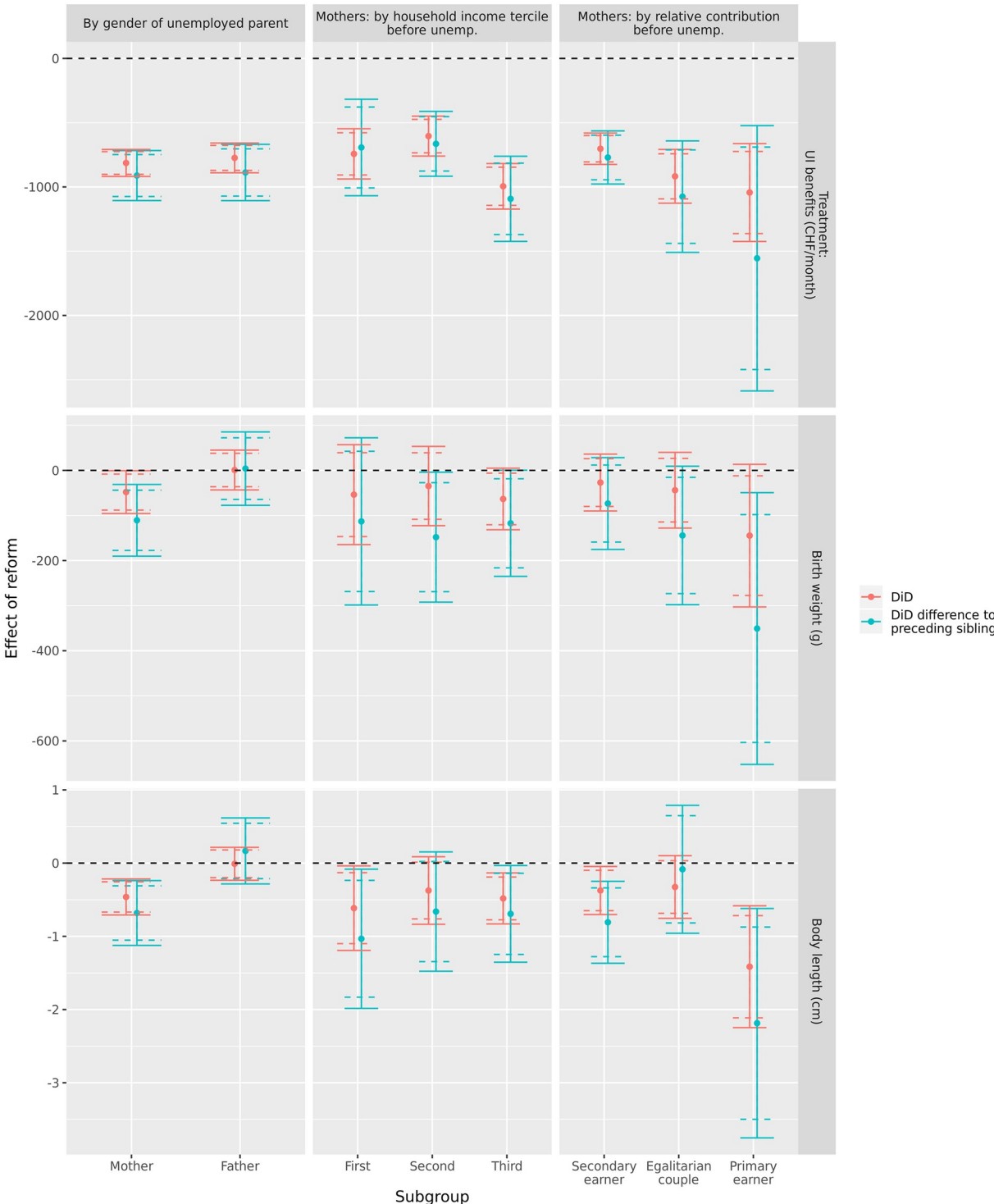

**Fig 1. Effect of reform on UI benefits (treatment) average birth weight, and body length.** Difference-in-differences estimates and 95% confidence intervals (dotted line: 90% confidence intervals), by gender of unemployed parent, and then for mothers by pre-unemployment household income, and relative income contribution. UI benefits measured in second year after onset of unemployment.

unemployment are strong enough to affect the health of *in utero* children of unemployed parents [8–10]. By allowing for better maternal nutrition, access to healthcare and by reducing stress, cash transfers have been shown to buffer such effects and to improve birth outcomes [20, 21]. However, prior studies focused on programs targeting the poorest in the US, which poses the question whether beneficial effects of cash transfers also apply to more general populations and to contexts with fewer health inequalities.

Exploiting quasi-random variation in unemployment benefits due to a Swiss policy reform and merged parent-child register data, this study provides evidence on the effect of cash transfers on newborn health when benefits are less targeted and when the population has more equitable access to healthcare and generally better birth outcomes [11, 26]. While the reform had no effect on children whose fathers were affected by the reform, it reduced average birth weight and body length of newborns whose mothers were affected by the reform.

The absolute size of the effect is significant. The estimated average effects of the reform on birth weight (~ -80g) are around half of the average effect of unemployment [cf. 8] and around 25% of the effect of smoking 6–10 cigarettes per day [36]. Previous studies suggested that reductions in birth weight of this magnitude would be associated with a reduction in earnings of 1.36% when these children are young adults [35].

The reform had a similar effect on children in different household income strata. Thus, given that the reform also affected mothers who did not depend on UI benefit income to be able to afford adequate healthcare or healthy nutrition, we assume that these were not the main mechanisms underlying the effect of the reform. This conclusion is not too surprising, given Switzerland's relatively generous healthcare and social assistance schemes that reduce the role of household purchasing power in ensuring a healthy pregnancy. Hence, the main mechanism through which the reform affected birth outcomes in Switzerland was likely stress during gestation and, possibly, stress-induced negative health behaviors such as smoking and drinking. This would also align with the clearly increased effects among mothers that were primary earners before job loss. Given their role in the household, it seems likely that these women felt more job search stress due to shortened benefits.

Our data do not allow us to directly test the stress mechanism underlying the reform's negative effect on birth outcomes. That said, given the Swiss context and the subgroup analysis finding the strongest impacts among unemployed primary earning women (and no differences by household income), we would argue that our results can be seen as additional evidence on the role of stress for birth outcomes [12, 15, 40, 41].

These results also speak to the general literature on economic shocks, stress and health [27, 28]. On average, job loss does not provoke immediate health reactions that can be (easily) captured with indicators of overall self-rated health or chronic conditions [1, 3, 5–7]. However, our results suggest that financial difficulties after job loss increases stress for unemployed pregnant women with somatic consequences for their children. Pregnant women are a special population with reduced abilities to react to less benefits by finding a job quicker and are thus more dependent on benefit income. This makes it less clear whether the stress mechanism suggested by our results can be generalized to the general population of unemployed individuals. That said, it is possible that other groups with similar job search restrictions (e.g. those with pre-existing health conditions or those with child care duties) might have similar stress reactions to reductions in benefit generosity. Our results thus encourages future research to consider more direct indicators of stress, e.g. measures of blood cortisol [cf. 45], to better understand the effect unemployment and unemployment benefits on stress.

Our study also has important policy implications. Even among relatively affluent households in a context with a generous social safety net and universal healthcare, reductions in unemployment benefits can impact newborn health. Cash transfers thus play a crucial role in

protecting newborn health not only among the poor in the US [17, 20], but also among large segments of the populations in contexts with less health inequalities [11, 26]. These results underscore the importance of special exemptions for pregnant women in unemployment insurance (such as removal of the job search requirement after gestational week 28 in Switzerland) and suggest caution when reducing unemployment insurance and other cash transfers.

## Supporting information

**S1 Table. Regression coefficients underlying Fig 1 (left-hand panel), estimates for children with an unemployed mother.**
(DOCX)

**S2 Table. Birth outcomes of children of unemployed mothers, controls and treated, pre and post-reform means, difference-in-differences estimates, and 95% confidence intervals of the effect of the reform.** 8 months unemployment duration instead of 9.
(DOCX)

**S3 Table. Birth outcomes of children of unemployed mothers, controls and treated, pre and post-reform means, difference-in-differences estimates, and 95% confidence intervals of the effect of the reform.** 10 months unemployment duration instead of 9.
(DOCX)

## Acknowledgments

We thank the Swiss Federal Statistical Office for providing the data, Leen Vandecasteele, Ursina Kuhn, René Rüegg, editors and referees from PLOS ONE and Journal of Health Economics as well as research workshop participants at Bern University of Applied Sciences and University of Lausanne for comments on prior versions of this manuscript.

## Author Contributions

**Conceptualization:** Dorian Kessler, Debra Hevenstone.

**Data curation:** Dorian Kessler.

**Formal analysis:** Dorian Kessler.

**Funding acquisition:** Dorian Kessler, Debra Hevenstone.

**Investigation:** Dorian Kessler.

**Methodology:** Dorian Kessler, Debra Hevenstone.

**Project administration:** Debra Hevenstone.

**Resources:** Debra Hevenstone.

**Supervision:** Debra Hevenstone.

**Visualization:** Dorian Kessler.

**Writing – original draft:** Dorian Kessler, Debra Hevenstone.

**Writing – review & editing:** Dorian Kessler, Debra Hevenstone.

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
