## [Decision Letter · Decision Letter 0]

28 Dec 2021

PONE-D-21-36369The impact of unemployment benefits on birth outcomes: Quasi-experimental evidence from European linked register dataPLOS ONE

Dear Dr. Kessler,

Thank you for submitting your manuscript to PLOS ONE. After careful consideration, we feel that it has merit but does not fully meet PLOS ONE’s publication criteria as it currently stands. Therefore, we invite you to submit a revised version of the manuscript that addresses the points raised during the review process. The revised version should address all comments. There is a large literature on the complex relationship between unemployment and health (see https://doi.org/10.1002/hec.1361).

We look forward to receiving your revised manuscript.

Kind regards,

Petri Böckerman

Academic Editor

PLOS ONE

Journal Requirements:

(We thank the Swiss National Science Foundation for funding for this project (grant nr. 176371), the Swiss Federal Statistical Office for providing the data, Leen Vandecasteele, Ursina Kuhn, René Rüegg, editors and anonymous referees from Journal of Health Economics as well as research workshop participants at Bern University of Applied Sciences and University of Lausanne for comments on prior versions of this manuscript.)

(Data collection, processing, analysis, and writing of the manuscript were funded by the Swiss National Science Foundation (grant no. 176371).)

5. Please note that supplementary tables (should remain/ be uploaded) as separate "supporting information" files.

Reviewers' comments:

Reviewer's Responses to Questions

**Comments to the Author**

1. Is the manuscript technically sound, and do the data support the conclusions?

Reviewer #1: Yes

Reviewer #2: Yes

2. Has the statistical analysis been performed appropriately and rigorously? 

Reviewer #1: Yes

Reviewer #2: Yes

3. Have the authors made all data underlying the findings in their manuscript fully available?

Reviewer #1: Yes

Reviewer #2: Yes

4. Is the manuscript presented in an intelligible fashion and written in standard English?

Reviewer #1: Yes

Reviewer #2: Yes

5. Review Comments to the Author

Reviewer #1: The paper is very interesting! The choosen topic is very actual! The Authors tried to analyse a relationship between the unemployment and the social benefits.

I suggest to make some small modifications:

- Line 51. Page 3. The Authos wrote: "The current literature focuses on the US context and on social programs targeting the very poorest." Please add the citations of the current literature.

- What do you think about the birth outcomes? What kinf of factors can influence the birth outcomes, only the unemployment insurance? What about the mother general's health or age? Please add some new paragraphs!

- The methodology part is very clear!

Reviewer #2: It appears to be a sound paper. However, the current results have not been discussed in detail with respect to literature. Again, the authors should consider discussing the results to highlight consistencies and inconsistencies of their findings.

6. PLOS authors have the option to publish the peer review history of their article (what does this mean?). If published, this will include your full peer review and any attached files.

Reviewer #1: **Yes: **Katalin Liptak

Reviewer #2: No

---

## [Author Response · Author response to Decision Letter 0]

1 Feb 2022

Comment by editor Petri Böckerman

There is a large literature on the complex relationship between unemployment and health (see https://doi.org/10.1002/hec.1361).

As a main goal of unemployment insurance is the reduction of the adverse consequences of unemployment on well-being, we fully agree that the effects of unemployment on health is a natural starting point of our paper. In the introduction (p. 3), we therefore referenced papers on the effects of unemployment on mental health, physical health, and newborn health. We also referred to the relevance of our results for this literature in the conclusion section (p. 18).

Comments by Reviewer 1

Reviewer #1: The paper is very interesting! The chosen topic is very actual! The Authors

tried to analyse a relationship between the unemployment and the social benefits.

I suggest making some small modifications:

- Line 51. Page 3. The authors wrote: "The current literature focuses on the US context and

on social programs targeting the very poorest." Please add the citations of the current

literature.

We added a citation referencing the relevant studies.

- What do you think about the birth outcomes? What kind of factors can influence the birth

outcomes, only the unemployment insurance? What about the mother general's health or

age? Please add some new paragraphs!

Thank you! The paper was indeed lacking basic background information on the determinants of birth outcomes and possible mechanisms that link unemployment and unemployment insurance to birth outcomes. After the introduction (p. 5), we added a paragraph with an overview of the determinants of birthweight and empirical evidence of those determinants. We also added a paragraph on the mechanisms that link unemployment and fetal development which helps the reader to follow our paragraph on the effect of the UI reform on birth outcomes.

Comments by Reviewer 2

Reviewer #2: It appears to be a sound paper. However, the current results have not been

discussed in detail with respect to literature. Again, the authors should consider discussing

the results to highlight consistencies and inconsistencies of their findings.

We agree that in the prior version we did not pay enough attention to the implications of our results with respect to the literature. In the revision we first clarify the mechanisms underlying our results (lines 41-43; 101-119; 356-364), and second, link the paper to the literature on the determinants of birth outcomes (specifically the role of stress) and to the literature on economic shocks (specifically unemployment), stress and health (lines 90-91). In the conclusion section, we interpreted our results considering this literature, highlighting its contributions and limits to generalizability (366-383).

---

## [Decision Letter · Decision Letter 1]

14 Feb 2022

The impact of unemployment benefits on birth outcomes: Quasi-experimental evidence from European linked register data

PONE-D-21-36369R1

Dear Dr. Kessler,

We’re pleased to inform you that your manuscript has been judged scientifically suitable for publication and will be formally accepted for publication once it meets all outstanding technical requirements.

Kind regards,

Petri Böckerman

Academic Editor

PLOS ONE

Additional Editor Comments (optional):

Reviewers' comments:

Reviewer's Responses to Questions

**Comments to the Author**

1. If the authors have adequately addressed your comments raised in a previous round of review and you feel that this manuscript is now acceptable for publication, you may indicate that here to bypass the “Comments to the Author” section, enter your conflict of interest statement in the “Confidential to Editor” section, and submit your "Accept" recommendation.

Reviewer #1: All comments have been addressed

2. Is the manuscript technically sound, and do the data support the conclusions?

Reviewer #1: Yes

3. Has the statistical analysis been performed appropriately and rigorously? 

Reviewer #1: Yes

4. Have the authors made all data underlying the findings in their manuscript fully available?

Reviewer #1: Yes

5. Is the manuscript presented in an intelligible fashion and written in standard English?

Reviewer #1: Yes

6. Review Comments to the Author

Reviewer #1: The Authors made a lot of changes in the paper! They added new literatures and wrote amuch better informed literature review. They added a paragraph with an overview of the determinants of birthweight and empirical evidence of those determinants. They wrote about the mechanisms that link unemployment and fetal development. The paper is now suitable for publication.

7. PLOS authors have the option to publish the peer review history of their article (what does this mean?). If published, this will include your full peer review and any attached files.

Reviewer #1: **Yes: **Katalin Liptak

---

## [Editor Report · Acceptance letter]

21 Feb 2022

PONE-D-21-36369R1 

The impact of unemployment benefits on birth outcomes: Quasi-experimental evidence from European linked register data 

Dear Dr. Kessler:

I'm pleased to inform you that your manuscript has been deemed suitable for publication in PLOS ONE. Congratulations! Your manuscript is now with our production department. 

Kind regards, 

on behalf of

Professor Petri Böckerman 

Academic Editor

PLOS ONE